# Variation in a range of mTOR-related genes associates with intracranial volume and intellectual disability

M.R.F. Reijnders [1], M. Kousi[2], G.M. van Woerden[3], M. Klein [1], J. Bralten[1], G.M.S. Mancini[4], T. van Essen[5], M. Proietti-Onori[3], E.E.J. Smeets[6], M. van Gastel[7], A.P.A. Stegmann[6], S.J.C. Stevens[6], S.H. Lelieveld[8], C. Gilissen [1], R. Pfundt[1], P.L. Tan[2], T. Kleefstra[1], B. Franke [1,9], Y. Elgersma[3], N. Katsanis[2] & H.G. Brunner[1,6]

De novo mutations in specific mTOR pathway genes cause brain overgrowth in the context of intellectual disability (ID). By analyzing 101 mMTOR-related genes in a large ID patient cohort and two independent population cohorts, we show that these genes modulate brain growth in health and disease. We report the mTOR activator gene *RHEB* as an ID gene that is associated with megalencephaly when mutated. Functional testing of mutant *RHEB* in vertebrate animal models indicates pathway hyperactivation with a concomitant increase in cell and head size, aberrant neuronal migration, and induction of seizures, concordant with the human phenotype. This study reveals that tight control of brain volume is exerted through a large community of mTOR-related genes. Human brain volume can be altered, by either rare disruptive events causing hyperactivation of the pathway, or through the collective effects of common alleles.

[1] Department of Human Genetics, Radboud University Medical Center, Donders Institute for Brain, Cognition and Behaviour, Nijmegen 6500 GA, The Netherlands. [2] Center for Human Disease Modeling, Duke University, Durham, NC 27701, USA. [3] Department of Neuroscience and ENCORE Expertise Center for Neurodevelopmental Disorders, Erasmus University Medical Center, 3015 CN Rotterdam, The Netherlands. [4] Department of Clinical Genetics, Erasmus MC, Sophia Children's Hospital, 3000 CA Rotterdam, The Netherlands. [5] Department of Genetics, University of Groningen, University Medical Center of Groningen, 9700 RB Groningen, The Netherlands. [6] Department of Clinical Genetics and School for Oncology & Developmental Biology (GROW), Maastricht University Medical Center, 6202 AZ Maastricht, The Netherlands. [7] Department of Medical Care, SWZ zorg, 5691 AG Son, The Netherlands. [8] Department of Human Genetics, Radboud Institute for Molecular Life Sciences, Radboud University Medical Center, 6500 GA Nijmegen, The Netherlands. [9] Department of Psychiatry, Donders Institute for Brain, Cognition and Behaviour, Radboud University Medical Center, 6500 GA Nijmegen, The Netherlands. M.R.F. Reijnders, M. Kousi, G.M. van Woerden, Y. Elgersma, N. Katsanis and H.G. Brunner contributed equally to this work. Correspondence and requests for materials should be addressed to H.G.B. (email: han.brunner@radboudumc.nl)

Many aspects of brain homeostasis, among which are measures of total brain volume, are highly heritable[1]. Genome-wide association studies (GWAS) of brain volume have shown a polygenic architecture in the general population, with individual common genetic variants explaining <1% of phenotypic variance[2]. Neurodevelopmental disorders, such as intellectual disability (ID) and autism spectrum disorder (ASD), have been associated with significant brain overgrowth. In ID, up to 6% of the patients are macrocephalic[3]. One of the key regulators of normal brain development is the evolutionarily conserved Ser/Thr protein kinase Mammalian Target Of Rapamycin (*MTOR*). The role of the mTOR pathway in brain development and function has been intensively studied both in vitro and in vivo using different mouse models. In these models, mutations in either the downstream effectors of mTOR, or the most important upstream regulators of mTOR, such as Ras homolog enriched in brain (Rheb), tuberous sclerosis 1 (Tsc1), and Tsc2, have been tested[4, 5]. Collectively, all studies provide strong evidence that proper mTOR signaling is involved in key aspects of brain development, such as neuronal progenitor maintenance and differentiation (including regulation of neuronal polarity, soma size and neurite outgrowth) and neuronal migration[6–17]. In the mature brain, mTOR is an important regulator of synapse formation and synaptic function[18–21], in particular through its role in regulating protein translation and elongation[22–25]. Not surprisingly, hyperactivity of the mTOR pathway in mice can lead to a myriad of phenotypes such as macrocephaly, seizures, and behavioral abnormalities[6, 26–31].

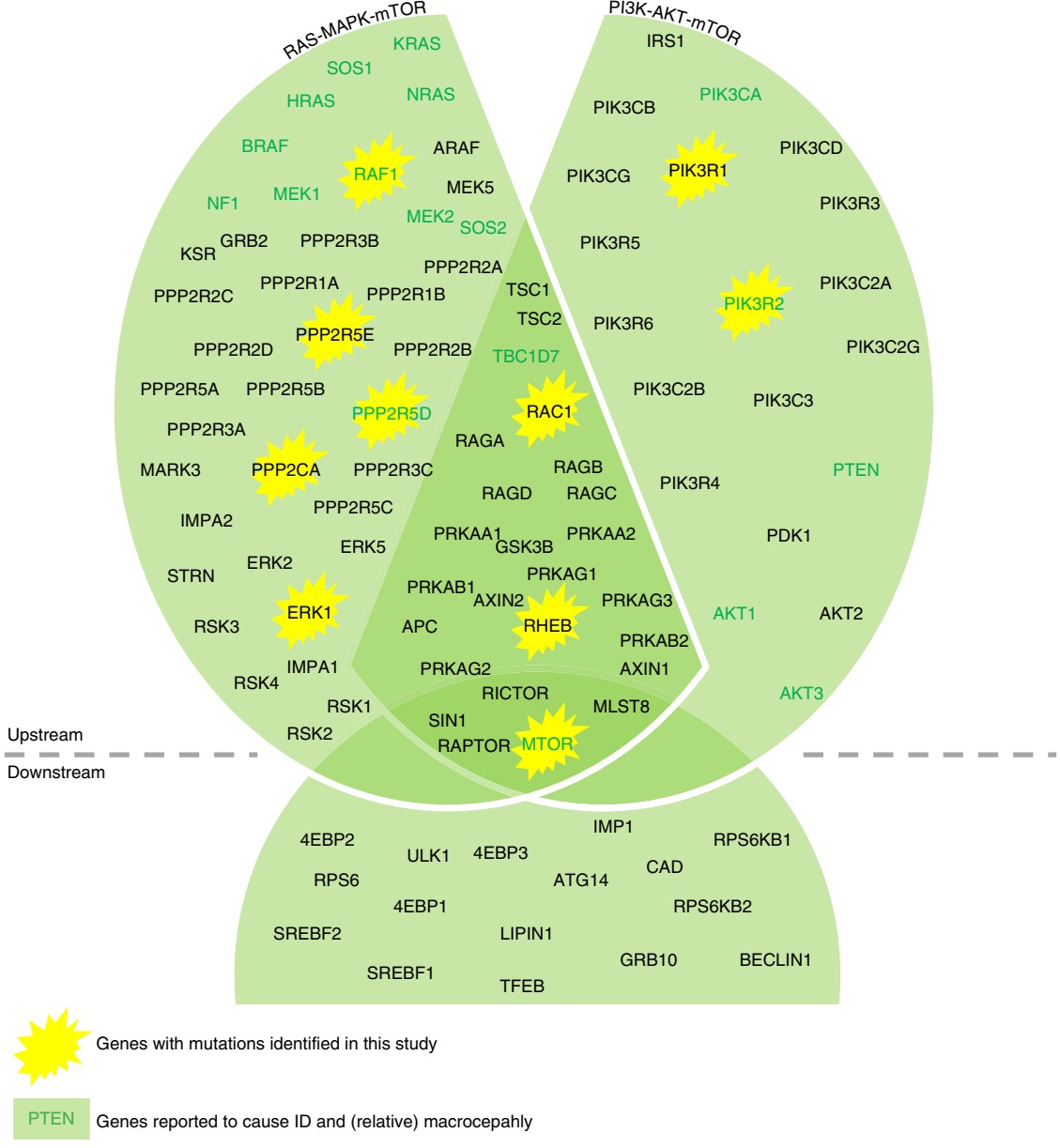

**Fig. 1** Schematic overview of selected mTOR-related genes. Schematic representation of the genes (*n* = 101) included in our mTOR-related gene-set based on three different authoritative publications[59-61]. Both proteins acting upstream of mTOR and proteins acting downstream of mTOR were included. Additionally, we subdivided the total set in two subsets: the RAS-MAPK-mTOR subset and the PI3K-AKT-mTOR subset. In both subsets, downstream genes are included as well. Genes in which we identified de novo mutations in this study were marked with a yellow star, and genes previously reported to cause ID and (relative) macrocephaly are shown in bolt and dark green

**Table 1 Identified mutations with bonferroni corrected *p*-value and occipital frontal circumference (OFC) of patients with de novo mutations in mTOR-related genes**

| Patient key | Gene | c.DNA | Protein change | Known ID gene | OFC | Bonferroni-corrected *p*-value |
|---|---|---|---|---|---|---|
| 1 | RHEB | c.202 T>C | p.(S68P) | No | >+2.5 SD | 4.514e−03* |
| 2 | RHEB | c.110 C > T | p.(P37L) | No | >+2.5 SD | |
| 3 | RHEB | c.110 C > T | p.(P37L) | No | >+2.5 SD | |
| 4 | RAC1 | c.53 G > A | p.(C18Y) | No | <−2.5 SD | 7.657e−03 |
| 5 | RAC1 | c.116 A > G | p.(N39S) | No | <−2.5 SD | |
| 6 | PPP2R5E | c.605 T > G | p.(V202G) | No | No data | 1 |
| 7 | PPP2CA | c.882dup | p.(R295*) | No | No data | 1.696e−02 |
| 8 | PPP2CA | c.572 A > G | p.(H191R) | No | >+2.5 SD | |
| 9 | ERK1 | c.569 T > C | p.(I190T) | No | <−2.5 SD | 1 |
| 10 | PIK3R1 | c.1359 C > G | p.(N453K) | Yes | Normal | 7.662e−02 |
| 11 | PIK3R1 | c.1692 C > G | p.(N564K) | Yes | >+2.5 SD | |
| 12 | PIK3R2 | c.1117 G > A | p.(G373R) | Yes | >+2.5 SD | 1 |
| 13 | RAF1 | c.1082 G > C | p.(G361A) | Yes | Normal | 1 |
| 14 | PPP2R5D | c.1258 G > A | p.(E420K) | Yes | No data | 7.832e−04 |
| 15 | PPP2R5D | c.598 G > A | p.(E200K) | Yes | >+2.5 SD | |
| 16 | PPP2R5D | c.592 G > A | p.(E198K) | Yes | >+2.5 SD | |
| 17 | MTOR | c.4555 G > A | p.(A1519T) | Yes | >+2.5 SD | 1 |

OFC occipital frontal circumference; SD standard deviation
*In the statistical enrichment analysis, the RHEB p.(P37L) variant was considered as a single event

**Table 2 Number of patients with macrocephaly, normal OFC, and microcephaly**

| | Macrocephaly | Normal OFC | Microcephaly |
|---|---|---|---|
| Patients with clinical data (*n* = 732) | 47 | 580 | 105 |
| Patients with de novo mutation(s) (*n* = 553) | 35 | 442 | 76 |
| Patients with de novo mutation in mTOR-related gene (*n* = 14) | 9 | 2 | 3 |
| Patients with de novo mutation in gene not related to mTOR (*n* = 539) | 26 | 440 | 73 |

OFC occipital frontal circumference

In contrast, sustained downregulation of the mTOR pathway appears to have little effect on neuronal function and behavior[32]. Findings that the epilepsy and behavioral deficits in mice can be rescued by mTOR inhibitors, offers a broad therapeutic window in which patients can potentially be treated. Indeed, recent studies indicated that mTOR inhibition is a promising treatment for epilepsy in tuberous sclerosis complex (TSC) patients[33–35].

Given the large body of evidence implying mTOR function in key aspects of brain development, it is not surprising that hyperactivating, somatic, and germline mutations in components of the PI3K-AKT3-mTOR pathway have been linked with rare ID syndromes associated with (hemi)megalencephaly, focal cortical dysplasia, and epilepsy[38–41]. We were struck by the apparent recurrence of mTOR-related mutations in ID, the persistent co-morbid megalencephaly and the absence of studies investigating the overall contribution of the mTOR pathway to ID and brain growth. Considering this knowledge gap, we sought to identify deleterious germline mutations in mTOR-related genes, and assess their contribution to the development of ID and megalencephaly. Next, assuming that our findings are not only relevant to rare diseases such as ID, we hypothesized that the pathology of syndromic ID patients represents the extreme end of a more continuous contribution of the mTOR pathway to human brain development and neuroanatomical variance in the population. Our data indeed indicate that mTOR variation significantly contributes to megalencephaly in a large ID cohort and brain size in the population. Furthermore, we present that de novo mutations in a key regulator of mTOR, *RHEB*, causes severe ID, epilepsy and megalencephaly in humans. By functionally testing the *RHEB* mutations in vertebrate animal models, we show that the specific mutations cause hyperactivation of mTOR, with a concomitant increase in cell and head size, aberrant neuronal migration and induction of seizures, concordant with the human phenotype. The extent of mTOR activation likely affects brain volume in humans. In extreme cases, highly deleterious mutations can lead to profound pathology. For such patients, functional restoration of the pathway through treatment with selective mTOR inhibitors might be of direct clinical utility.

## Results

**mTOR-related mutations are associated with macrocephaly**. To assess the overall burden of mTOR defects to ID, we performed whole-exome sequencing (WES) in a cohort of 826 patients with ID cataloguing de novo mutations (Supplementary Data 1) in a set of 101 mTOR-related genes (Supplementary Data 2, Fig. 1). We identified 17 de novo mutations affecting 10 different mTOR-related genes, providing a possible genetic diagnosis in 2.1% of our cohort. Five of the identified genes were known ID genes (*PIK3R1*, *PIK3R2*, *RAF1*, *PPP2R5D*, *MTOR*) and five (*RHEB*, *RAC1*, *PPP2R5E*, *PPP2CA*, *ERK1*) were not associated with ID previously (Fig. 1, Table 1, Supplementary Data 3). Three of the five novel genes (*RHEB*, *RAC1*, and *PPP2CA*) showed a significant enrichment for de novo mutations in our patient cohort (Table 1, Supplementary Table 1). Combining the gene-specific mutation rates of all individual mTOR-related genes, we found a significant enrichment for de novo mutations in mTOR-related genes (*p* = 3.50e−04) (Supplementary Table 1). Additionally, we found significant spatial clustering of de novo missense variants for a single gene (PPP2R5D:

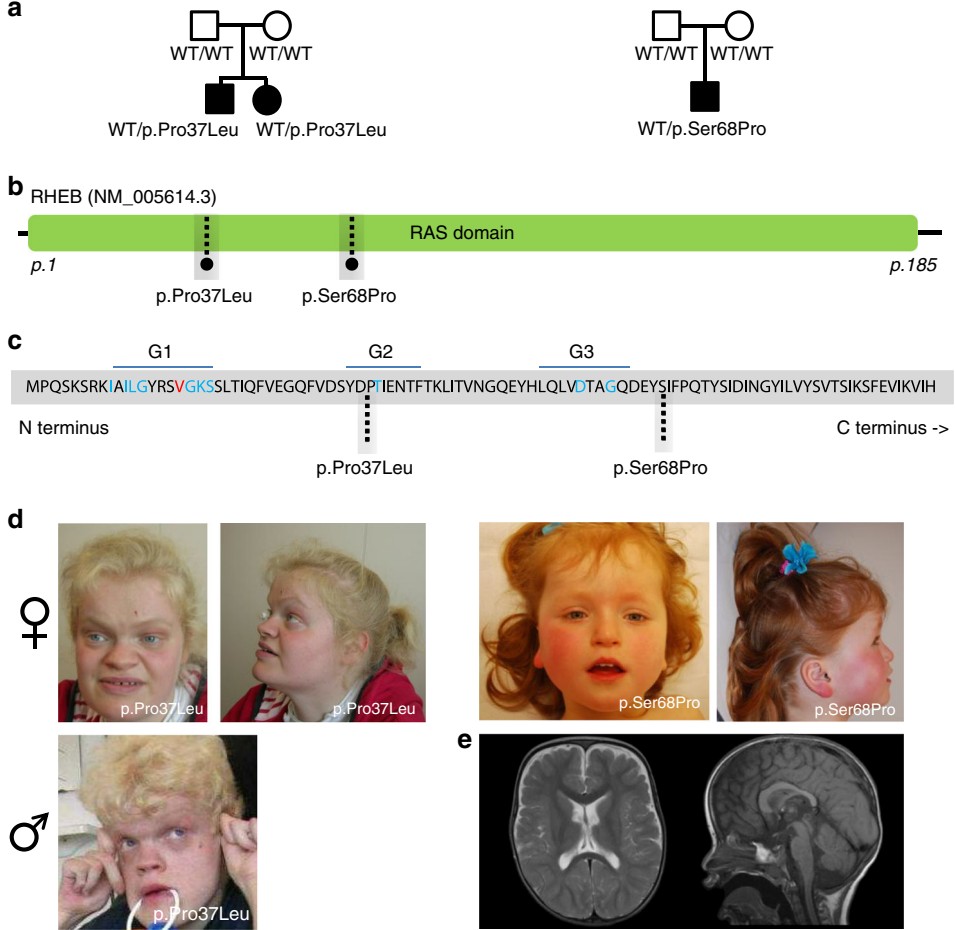

**Fig. 2** De novo mutations in *RHEB* cause an ID syndrome associated with megalencephaly. **a** Pedigree structure, disease status and genotype information for patients with changes in *RHEB*. **b** Schematic of the RHEB protein and the RAS domain. **c** Schematic of the N-terminal portion of the human RHEB protein. G-box residues characteristic of RAS superfamily proteins' are shown in blue; highly conserved residue conserved in 90% of the RAS superfamily members are shown in red. Dotted lines are showing the amino-acid residues mutated in patients described in the context of this study. **d** Photographs of the probands carrying de novo variants in *RHEB*. **e** MRI images (left: axial, T2-weighted; right: sagittal midline, T1-weighted) of the proband (age 1 year, 9 months) carrying the de novo *RHEB*p.S68P variant, showing macrocephaly, megalencephaly, broad frontal lobes, mild dilatation of lateral ventricles, large rostrum of corpus callosum and mild hypoplasia of the lower cerebellar vermis. No cortical malformations have been observed

$p < 1e-07$; permutation test) and a general pattern of spatial clustering across the five genes with recurrent de novo missense variants ($p = 0.0057$, Fisher's combined probability test; Supplementary Table 2).

To investigate the contribution of mTOR-related mutations on brain overgrowth, we performed a literature analysis of the 101 mTOR genes. This search showed that 23 genes had been previously reported to cause syndromic ID, with the majority (18/23; 78%) being associated with varying degrees of macrocephaly or relative macrocephaly (Supplementary Table 3). Motivated by this observation, we collected occipital frontal circumference (OFC) data from 732/826 patients (Supplementary Data 4). Macrocephaly was present in 6% of patients in our cohort (47/732 ID patients), a rate comparable to previous reports from an independent cohort[3]. De novo mutations were identified in 76% of our cohort (553/732 patients; Table 2). Among the 35 patients presenting with ID, macrocephaly, and a de novo mutation, we found a significant enrichment ($p = 9.084e-09$) for de novo mutations within genes of the mTOR pathway (9/14) compared to genes that operate in mTOR independent pathways (26/539) (Table 2). In contrast, microcephaly was not enriched among patients with de novo mutations in mTOR-related genes ($p = 0.4228$).

**mTOR pathway contributes to intracranial volume.** Driven by the high frequency of brain overgrowth described in the literature and the strong enrichment of macrocephaly in patients with mutations in mTOR-related genes in our cohort, we tested our set of 101 mTOR pathway genes for an association with intracranial volume (ICV) in the general population (Fig. 1, Supplementary Table 2). The final data set contained 76,746 SNPs in 96 autosomal genes (data were unavailable for X-chromosomal *ARAF*, *RPS6KA3*, *RPS6KA6*, *RRAGB*, and *PPP2R3B*). Using the ENIGMA2 data set ($n = 13,171$) we found a significant association of the entire mTOR gene set with ICV for the self-contained test ($p_{self-contained} = 0.0029088$) and a suggestive association for the competitive test ($p_{competitive} = 0.054742$). Data from the CHARGE consortium ($n = 12,803$) similarly revealed a significant association of the mTOR gene set with ICV for the self-contained test, but not for the competitive test ($p_{self-contained} = 0.00076589$ and $p_{competitive} = 0.22105$, respectively). Meta-analysis of the two data sets, confirmed the significant association of the mTOR gene set with ICV both for self-contained and competitive tests ($p_{self-contained} = 1.3895e-05$, $p_{competitive} = 0.025764$). Post hoc testing of the two major branches of the mTOR pathway separately (RAS-MAPK-mTOR, 76 genes; PI3K-AKT-mTOR, 60 genes; Fig. 1, Supplementary Table 4) showed stronger association

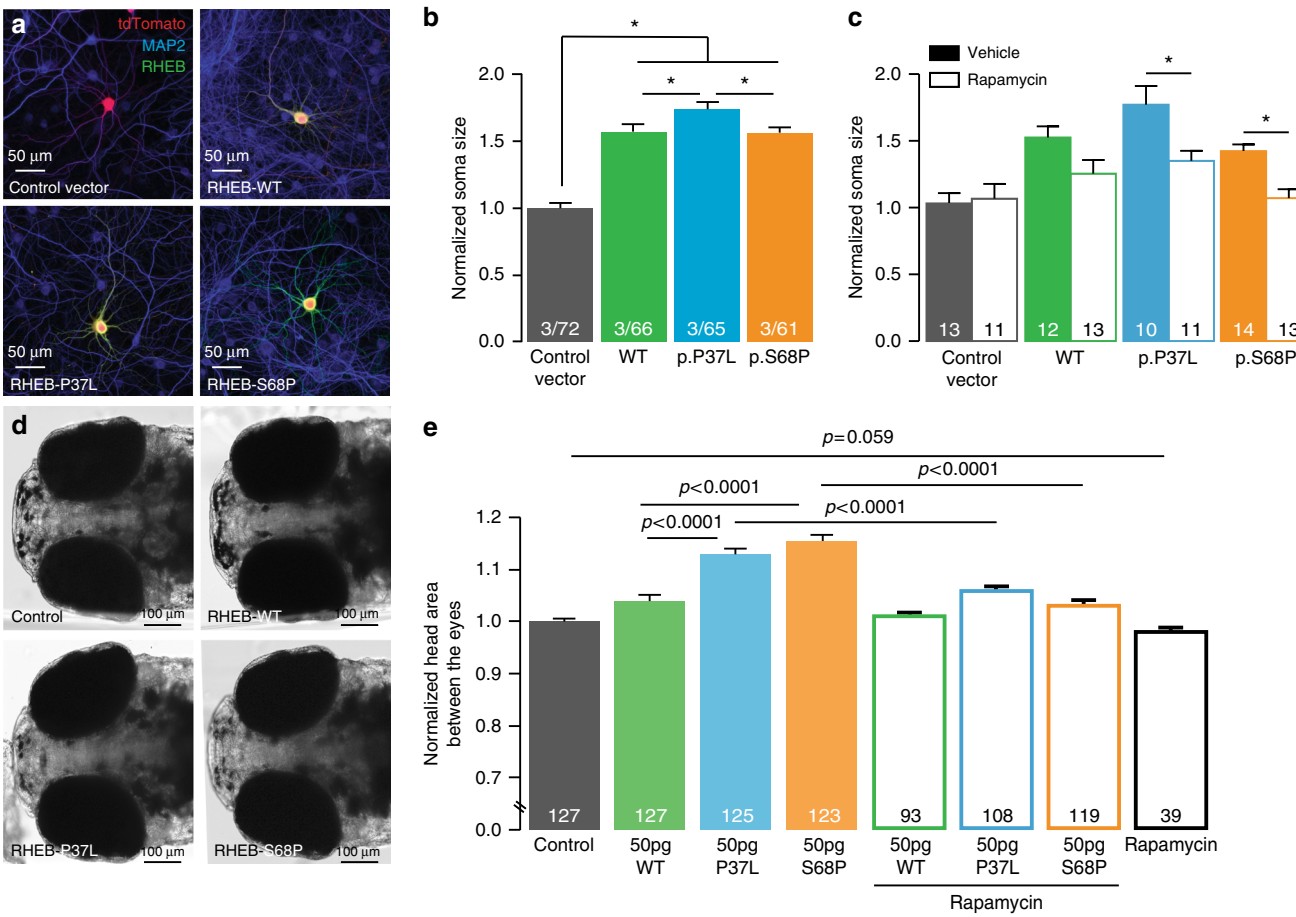

**Fig. 3** De novo mutations in *RHEB* increase soma size and headsize, phenotypes that can be rescued by rapamycin. **a** Representative confocal images of hippocampal neurons transfected with control vector, RHEB-WT, RHEBp.P37L or RHEBp.S68P. **b** Summary bar graph of soma size measured for each condition and normalized to the control vector. **c** Summary bar graph of soma size measured with and without rapamycin. Data are presented as mean ± SEM. Numbers depicted in the bar graph in **b** represent the number of independent cultures/total number of cells analyzed and in **c** number of cells analysed. Statistical significance was assessed by one-way ANOVA followed by Tukey's post hoc test (*$p < 0.01$). **d** Dorsal view of 5 dpf control and overexpressant zebrafish larvae. For each experiment, embryos were injected with either WT or mutant *RHEB* human mRNA message. The area between the eyes was measured for every embryo, to produce a quantitative score. **e** Bar graph showing the quantification of the headsize phenotype in control embryos and embryos injected with either WT or mutant human *RHEB* with and without rapamycin. The graph represents cumulative plotted experiments across three biological replicas. Statistical analyses were performed by Student's *t*-test

of PI3K-AKT-mTOR than RAS-MAPK-mTOR with ICV (PI3K-AKT-mTOR: $p_{self-contained} = 0.00092471$, $p_{competitive} = 0.0079133$; RAS-MAPK-mTOR: $p_{self-contained} = 2.2885e-07$, $p_{competitive} = 0.068983$). The role of the PI3K-AKT-mTOR pathway in volumetric variation of the brain was further strengthened by testing the previously described Reactome_PI3K_AKT_activation gene-set[42] (35 genes; $p_{self-contained} = 3.8649e-13$; $p_{competitive} = 0.00028957$; Supplementary Table 5). Not all 96 genes in the gene-set analysis showed significant association with ICV individually. The most strongly associated individual gene was *AKT3* ($P = 2.22E-05$) and in total, 18 genes of the mTOR gene set including *APC* ($P = 0.00042$), and the new ID gene *RHEB* ($P = 0.0041$), showed nominally significant association with ICV (Supplementary Fig. 1; Supplementary Table 6).

**RHEB mutations cause increased neuronal cell and head size.** Two of the three individuals with de novo *RHEB* mutations were siblings and carried the same heterozygous p.(Pro37Leu) mutation, while a sporadic individual carried the p.(Ser68Pro) allele. The p.(Pro37Leu) mutation was not identified in either parent, suggesting parental gonadal mosaicism (Fig. 2a). The

*RHEB* mutations are located in the RAS domain (Fig. 2b, c) and are absent from ExAC, EVS, or our internal clinical exome databases. All three individuals (Fig. 2d) with de novo *RHEB* mutations had short stature (−2 to −3 SD) and early brain overgrowth with pronounced macrocephaly during childhood (+2.5/+3 SD). They had severe to profound ID with hypotonia, as well as autism spectrum disorder. Two of three individuals were reported to have epilepsy. No epileptic episodes were noted for the third patient, but EEG recordings were suggestive of epileptic discharges (Supplementary Note, Supplementary Table 7). Brain magnetic resonance imaging (MRI) evaluation of the patient with the p.(Ser68Pro) allele, confirmed megalencephaly with broad frontal lobes and mild dilatation of the lateral ventricles. The MRI scan further showed a thickened rostrum of the corpus callosum and small splenium, and mild hypoplasia of the lower cerebellar vermis (Fig. 2e).

We selected the *RHEB* mutations to obtain experimental evidence for our hypothesis that de novo changes in mTOR-related genes are likely due to a gain-of-function mechanism, resulting in hyperactivation of mTOR, as previously shown for other syndromic neurodevelopmental cases associated with macrocephaly. We first tested in vitro whether the *RHEB* de

novo changes have an impact on overall mTOR activity levels. Given that mTORC1 regulates cell size[11, 43], we used primary hippocampal neuron soma size as a readout to assess differences between RHEB-WT overexpressing vs. RHEBp.P37L and RHEBp.S68P overexpressing neurons. A significant increase in soma size was detected already in *RHEB*-WT transfected neurons, suggesting that *RHEB* is a highly dosage sensitive gene, likely causing hyperactivation of the mTOR pathway[44]. Overexpression of the RHEB mutant proteins caused an increase in soma size, confirming that these mutations do not cause a loss of function (Fig. 3a, b, one-way ANOVA, $p < 0.0001$, $F(3,260) = 50.35$; control vector vs. RHEB-WT: $p < 0.0001$; control vector vs. RHEBp.P37L: $p < 0.0001$; control vector vs. RHEBp.S68P: $p < 0.0001$ by Tukey's multiple comparisons test). Notably, overexpression of RHEBp.P37L had the strongest effect inducing a significantly pronounced increase in soma size compared to RHEB-WT ($p < 0.05$) and RHEBp.S68P ($p < 0.05$).

We next sought to evaluate the relevance of these variants in the development of neuroanatomical phenotypes in a developing zebrafish in vivo model. Toward this, we identified the sole zebrafish *rheb* ortholog (96% similarity, 91% identity). First, we corroborated that the variants identified are not acting through a loss of function mechanism by generating a CRISPR-Cas9 system to introduce deletions. Assessment of head size in mosaic F0 embryos injected with a guideRNA against exon 3 showed microcephaly in two biological replicates, which was opposite to the phenotype observed in the patients (Supplementary Fig. 2). We next evaluated the effect of the *rheb* alleles on head size under a gain of function and mTOR hyperactivating paradigm, as suggested through our in vitro studies. To test this hypothesis, we injected human WT or mutant *RHEB* mRNA into 1- to 4-cell stage zebrafish embryos. Expression of WT human *RHEB* induced a significant increase in the headsize area of 5 dpf larvae ($p = 0.0013$). Overexpression of either *RHEB*p.P37L or p.S68P, also resulted in significantly increased headsize, reminiscent of the megalencephaly seen in our patients ($p < 0.0001$ for either mutant allele when compared to WT *RHEB*; Fig. 3d, e). This finding was reproducible across three independent biological replicates.

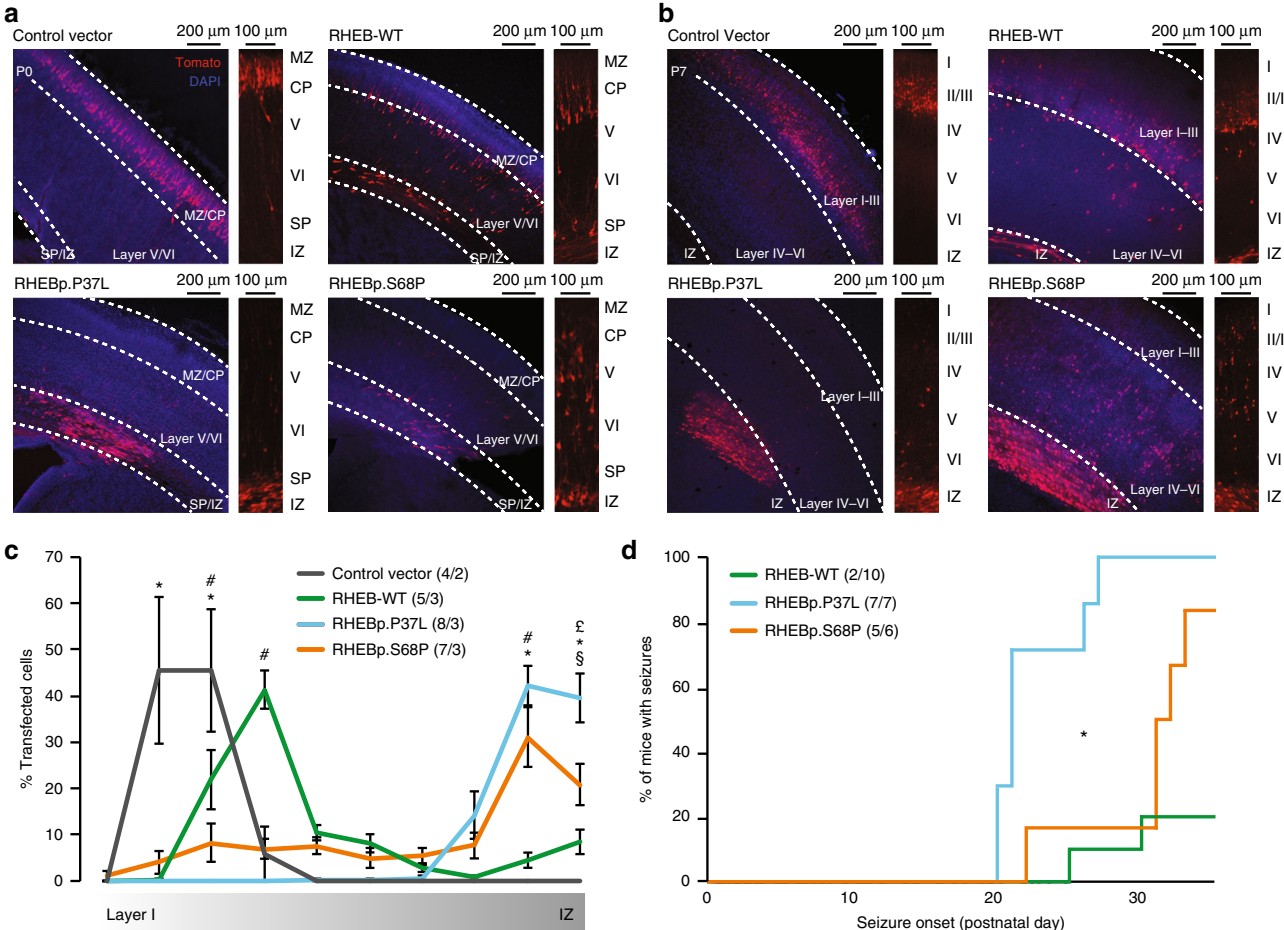

**Fig. 4** Overexpression of RHEB mutants in vivo causes deficits in neuronal migration and seizures in mouse. **a**, **b** Representative images of E14.5 in utero electroporated P0 brains (**a**) or P7 brains (**b**), with an enlargement showing the migratory pathway of the transfected cells (tdTomato+) from the intermediate zone (IZ) and subplate (SP) to the more superficial layers of the cortex (CP = cortical plate and MZ = marginal zone). **c** Quantification of the neuronal migration pattern observed in different conditions. Data are presented as mean ± SEM. Statistical significance was assessed by two-way repeated measure ANOVA followed by Bonferroni's post hoc test (for bins 2–4: *indicates significant difference between control vector and the different RHEB conditions ($p < 0.0001$); #indicates significant difference between the RHEB-WT and all other conditions ($p < 0.0001$); for bins 9 and 10: *indicates significant difference between control vector and RHEBp.P37L and RHEBp.S68P; #indicates significant difference between RHEB-WT and RHEBp.P37L and RHEBp.S68P ($p < 0.0001$)); §indicates significant difference between RHEB-WT and RHEBp.P37L ($p < 0.0001$); £indicates significant difference between RHEBp.P37L and RHEBp.S68P ($p < 0.001$). **d** Kaplan–Meier graph representing onset of tonic–clonic seizures in successfully targeted mice. The insert legends of the graph show $N_{pictures}/N_{mice}$ (**c**) or $N_{seizure}/N_{total}$ (**d**)

**Rapamycin rescues neuronal soma and head size defects.** Antagonists of the mTOR pathway, such as rapamycin, can ameliorate some of neurological deficits associated with mTOR hyperactivity[33–35, 36, 37]. To evaluate whether this is true for the *RHEB* activating mutations described here, we treated the neurons with 20 nM rapamycin or vehicle 1 day after transfection for 3 days and assessed neuronal soma size. We found that while the soma size of RHEB-WT overexpressing neurons nominally decreased, a statistically significant reduction of neuronal soma size was observed for both RHEBp.P37L and RHEBp.S68P and a trend in the same direction was seen for RHEB-WT (Fig. 3c, two-way ANOVA, effect of treatment $p < 0.0001$, $F(1,89) = 16.29$; RHEBp.P37L vehicle vs. RHEBp.P37L Rapamycin $p < 0.01$; RHEBp.S68P vehicle vs. RHEBp.S68P Rapamycin: $p < 0.05$; RHEB-WT vehicle vs. RHEB-WT Rapamycin: $p = 0.1$; by Bonferroni's multiple comparisons test). Taken together, these data show that overexpression of either wild-type or mutant *RHEB* induces an mTOR-dependent increase in soma size and that this phenotype can be rescued through the administration of the known mTOR antagonist rapamycin. Grounded on the in vitro observations, we next explored the possibility of rapamycin serving as a putative therapeutic agent in vivo. Toward this, we co-injected rapamycin together with WT or mutant *RHEB* mRNA in zebrafish embryos and we compared the embryos' head size at 5 dpf. Rapamycin alone did not induce any appreciable neuroanatomical pathologies, or indeed any other overt morphological phenotype(s) (Fig. 3e). In contrast, rapamycin sufficiently and reproducibly rescued the macrocephalic phenotype induced by both *RHEB*p.P37L and *RHEB*p.S68P, suggesting that suppression of mTOR hyperactivity might present a therapeutic target for disease amemlioration (Fig. 3e).

**RHEB mutations affect neuronal migration and induce seizures.** Previous studies have shown that mTOR signaling is not only involved in cell morphology and growth, but also plays a role in neuronal migration. Increased mTOR activity in vivo, induced either by overexpression of a constitutively active RHEB or by inactivating mutations in the *Tsc1* or *Tsc2* genes, two negative regulators of *RHEB*, causes neuronal migration defects[12–17]. We performed in utero electroporation at E14.5 to induce the in vivo overexpression of RHEB-WT, RHEBp.P37L, and RHEBp.S68P, and tested the effect of the *RHEB* mutations on neuronal migration in the still developing somatosensory cortex of P0 pups[45, 46]. Although in neuronal cultures overexpression of RHEB-WT and RHEB mutants increased soma size equally, the results obtained in vivo showed notable differences between these proteins. We observed that while cells transfected with the control vector efficiently migrated to the cortical plate (CP), cells transfected with RHEB-WT could be found in all the layers of the cortex (Fig. 4a). Strikingly, the majority of cells transfected with either RHEBp.P37L or RHEBp.S68P remained in the subplate (SP), indicating more severe migration deficits compared to RHEB-WT overexpression (Fig. 4a). At P7, when the cortical layers are more defined, the difference between RHEB-WT and RHEBp.P37L or RHEBp.S68P was even more striking (Fig. 4b). Analysis of the number of tdTomato-positive cells present in the different cortical layers showed a significant difference between the four different conditions (two-way repeated measure ANOVA, effect of interaction: $F(27,180) = 13.73$, $p < 0.0001$), consistent with our previous in vitro and in vivo studies that the mutations in *RHEB* are gain-of-function hyperactivating mutations. Consistent with our findings in primary neuronal cultures, post hoc analysis revealed that the RHEBp.P37L mutation yielded the strongest effects among evaluated conditions (Fig. 4c).

Neuronal migration deficits are often linked with seizures and ID[47]. Additionally, the link between an epileptogenic phenotype and hyperactivity of the mTOR pathway has been established from studies in both human and mice[28, 30, 48]. Interestingly, knockdown of the *TSC1* gene, a negative regulator of the mTOR pathway, in just a subset of cortical neurons reduces the threshold for seizure induction[12]. Careful monitoring of the in utero electroporated mice, revealed that overexpression of RHEB-WT, RHEBp.P37L, and RHEBp.S68P resulted in spontaneous tonic–clonic seizures starting at P20 (Supplementary Movie 1). Seizures were particularly common in mice expressing mutant RHEB: whereas 20% (2/10) of mice expressing RHEB1-WT developed epilepsy, all (7/7) mice expressing RHEBp.P37L and 83% (5/6) of mice expressing RHEBp.S68P developed spontaneous seizures (Fig. 4d). Consistent with our findings from primary neuronal cultures and neuronal migration following in utero electroporation, the RHEBp.P37L allele was shown to have the strongest effect, as the mice expressing this allele also showed a significantly earlier onset of seizures (log-rank (Mantel–Cox) $p < 0.01$ compared to RHEBp.S68P; Fig. 4d). Taken together, our in vivo results further corroborate the conclusion that the missense mutations in *RHEB* act as dominant activating mutations.

## Discussion

Here we studied the contribution of mTOR-related genes to ID and brain overgrowth in 826 ID patients unselected for any other phenotypic features and found 17 germline de novo mutations in genes related to mTOR, providing a possible genetic diagnosis for 2.1% of our cohort. We show that genes encoding components of the mTOR pathway, contribute to rare and common alleles that impact brain volume and provide insight into neurodevelopmental processes mediated through mTOR hyperactivity and outlook to potential treatment options for a subset of patients with ID.

A significant fraction of patients harboring a de novo mutation in mTOR-related genes was observed to be macrocephalic. The link between mTOR mutations and ID and/or head size differences has already been established through numerous studies that identified genes such as *AKT3*, *PIK3CA*, *PPP2R5D*, and recently *MTOR* itself[38–40, 49]. In fact, from the 23 genes previously reported to cause ID among our gene-set of 101 mTOR-related genes, most (18/23, 78%) have been associated in the literature with macrocephaly or relative macrocephaly. Our study significantly extends these findings: of the 35 patients with macrocephaly in the complete cohort, 9 patients (26%) harbored a de novo mutation in mTOR-related genes. As such, genes in this pathway should be carefully evaluated in patients with ID and macrocephaly.

Motivated by the high frequency of brain overgrowth in previous reports of mTOR-related syndromes, and in patients with mutations in mTOR-related genes in our cohort, we reasoned that the highly penetrant activating alleles that we identified de novo might represent only a fraction of alleles associated with severe neurocognitive disorders, and that more common and less penetrant alleles might be associated with head and brain growth in the general population. Indeed, a combined analysis of common variants of all 96 autosomal genes in the mTOR-related gene-set showed significant association with ICV in two large imaging genetics samples from the CHARGE and ENIGMA consortia, confirming our initial hypothesis. Interestingly, the PI3K-AKT-activation pathway (35 genes from the reactome gene-set) was recently shown to be among the most strongly associated pathways for ICV in an enrichment analysis testing 671 Reactome gene-sets using the same cohorts[42]. Our

analyses support and expand this conclusion by testing a different, carefully selected gene-set (only 15 out of 96 genes overlapping). Taken together, our data support a model by which mTOR-related genes, including the newly discovered ID gene *RHEB*, contribute to variation in brain growth, through common and rare genetic variants, in health and disease. Our observations corroborate, how rare disorders can inform biological mechanisms underlying common traits in the general population.

There is ongoing debate on the precise genetic composition of gene-sets. Gene-set databases, such as KEGG, Ingenuity, and others, all differ in their coverage of specific biological pathways and their functional annotations. In line with this observation, the number of genes mapped to pathways may also vary greatly across the databases[50]. For mTOR-related gene-sets, inclusion in databases is incomplete, with key proteins and protein complexes such as *RAC1*, *RAG*, *MEK*, and *PP2A* missing. For this reason, we used three authorative reviews on mTOR signaling describing both upstream and downstream interactors of mTOR and then used additional evidence from the literature to subdivide various protein complexes into their constituent proteins and genes. Therefore, our selection of 101 mTOR-related genes might be incomplete and additional genes are likely to be involved in mTOR signaling. For this reason and because of limitation of our methods to detect reliably somatic mosaicism, a mechanism thought to be a significant contributor to mutation burden in this pathway[38–40], we postulate that the diagnostic rate within our cohort (2.1%) might represent the lower bound of the estimate.

In this study, we identified de novo mutations in both known ID genes (*PIK3R1*, *PIK3R2*, *RAF1*, *PPP2R5D*, *MTOR*) and novel candidate ID genes (*RHEB*, *RAC1*, *PPP2R5E*, *PPP2CA*, *ERK1*). For the most frequently mutated gene, *RHEB*, we show that hyperactivating mutations cause an ID syndrome with brain overgrowth and epilepsy. The finding that these mutations are hyperactivating, is in line with the observation that loss of RHEB activity does not result in overt neurological phenotypes in *Rheb* mutant mice[32]. Several mechanisms, such as increased proliferation, increased soma size and reduced apoptosis are known to have a role in the development of megalencephaly[51]. We observed a significant increase in soma size upon overexpressing WT and mutant RHEB alleles in vitro. Since RHEB is the canonical activator of mTOR, this finding is consistent with other reports that have highlighted mTOR as a main regulator of cell size[6, 11, 15, 52]. In vivo, we postulate that the increased soma size might represent one of the mechanisms through which macrocephaly occurs, as the zebrafish embryos injected with mutant *RHEB* were phenotypically concordant with the human patients. Further dissecting the pathomechanism(s) underlying *RHEB*-associated ID, we showed severe neuronal migration defects in mouse embryos electroporated with mutant *RHEB* and an increased incidence of epileptogenic activity postnatally. These findings are reminiscent to what has been observed for mutations of *MTOR* itself. Constitutive activation of mTORC1 causes enlarged neuronal somata in rodent neurons, and focal cortical expression of *MTOR* mutations has been reported to disrupt neuronal migration and to cause spontaneous seizures by in utero electroporation in mice[40, 41, 53]. This observation shows that activating mutations in different genes of the mTORC1 branch of the mTOR pathway act through convergent mechanisms and have similar phenotypic outcomes.

Based on these observations, we reasoned that patients with activating *RHEB* changes might be able to benefit from therapies that result in a reduction of mTOR activity, such as rapamycin. Indeed, we here showed that suppression of mTOR levels through the administration of the mTOR antagonist rapamycin can significantly and reproducibly prevent both the neuronal soma size phenotype in vitro and the macrocephalic phenotype in vivo.

Recent studies have reported successful implementation of mTOR inhibitor treatment in individuals with TSC-associated epilepsy and brain tumors[33–35]. In a conceptually similar paradigm, fibroblasts from a patient with an mTOR activating *PIK3CA* change, were treated successfully with the PI3K inhibitors wortmannin or LY294002, which abrogated the overactivation of the pathway[54]. It is premature to advocate the use of rapamycin in patients with ID and mutations in all mTOR-related genes, not least because of the potentially adverse effects induced by prolonged exposure to this agent[55]. However, we speculate that targeted administration of mTOR inhibitors (rapamycin, wortmannin, everolimus as well as currently emerging second-generation drugs), perhaps during critical postnatal neurodevelopmental windows, might be of significant benefit to patients. In that context, rapid molecular diagnosis in both known ID genes and candidate ID genes, would be a critical component of the treatment decision process.

In conclusion, our data show that a large number of mTOR-related genes together modulate human brain volume in the population. Severe disruption of such mTOR-related genes can cause intellectual disability and brain overgrowth, most likely through mTOR hyperactivation.

## Methods

**Subjects and mutation analysis**. We evaluated a cohort of 826 patients with ID, who had undergone diagnostic trio WES at Radboud University Medical Center (Radboudumc). We included 820 simplex patients described previously, as well as three sib pairs excluded from the earlier study[56]. Diagnostic WES was approved by the medical ethics committee of the Radboud University Medical Center (Commissie Mensgebonden Onderzoek), Nijmegen, The Netherlands (registration number 2011-188).Written informed consent was obtained from all individuals or their legal guardians. We collected all available clinical information and performed deep phenotyping of individuals with a de novo mutation in *RHEB*. Consent for publication of photographs was obtained. Brain images were re-evaluated where available.

**Selection of mTOR-related genes**. We focused on the two well-described, convergent pathways in which mTOR acts as key regulator: the PI3K-AKT-mTOR pathway and the RAS-MAPK-mTOR pathway. We defined a list of 101 mTOR-related genes based on three authorative reviews on the mTOR regulators[57–59]. Protein complexes were mapped to single proteins and genes based on information available in the literature. The final list contains 101 mTOR-related genes: 96 map on autosomes and five map on the X-chromosome.

**Identification of mutations and collection of OFC data**. From our cohort of 826 patients with ID, we selected all de novo mutations that affect mTOR-related genes. All mutations were confirmed by Sanger sequencing. mTOR-related genes were considered to be known ID genes, if present in our recently published list containing over 1500 known ID genes[56]. We performed a literature search by querying Pubmed to investigate which of the known mTOR-related ID genes have been associated with large and small head size. Within our cohort of 826 patients, individuals were classified as microcephalic (OFC < −2.5 SD), macrocephalic (OFC > +2.5 SD), normocephalic (OFC between −2.5 SD and +2.5 SD) or unknown. We used Fisher's Exact test to calculate enrichment of macrocephaly in mTOR-related mutation carriers. The significance threshold was set at $p < 0.05$.

**Gene-based enrichment**. To assess whether mTOR-related genes were significantly enriched for functional de novo mutations in our cohort, we tested each of the 101 genes using a statistical model as described previously[56]. For this statistical enrichment analysis, the RHEB p.(P37L) variant was considered as one single event. Multiple testing correction was performed by the Bonferroni procedure based on 101 tested genes. Additionally, we tested whether the mTOR pathway as a whole was enriched for functional de novo mutations in our cohort by combining the gene-specific mutation rates of all individual genes in the pathway.

**Clustering analysis**. Clustering analysis was performed by generating the full cDNA for the respective RefSeq genes. To increase the statistical power of the spatial clustering of the recurrently mutated genes, we added de novo missense variants from the denovo-db[60] annotated by our in-house pipeline (Supplementary Data 5). The locations of observed de novo missense mutations were randomly sampled 100,000 times over the cDNA of the gene after which the distances (in base pairs) between the mutations were normalized for the total coding size of the respective gene. The geometric mean (the nth root of the product of n

numbers) of all mutation distances between the mutations was taken as a measure of clustering. A pseudocount (adding 1 to all distances and 1 to the gene size) was applied to avoid a mean distance of 0 when there were identical mutations. To assess overall clustering of the set of genes, we used Fisher's combined probability test to combine the 5 $p$-values of individual genes. To avoid a possible bias introduced by highly significant $p$-values (e.g., gene PPP2RD5), we calculated the combined $p$-value on deflated $p$-values where all values smaller than 0.05 were set to 0.05.

**ENIGMA and CHARGE study populations and data description**. This study reports data on 25,974 subjects of Caucasian ancestry from 46 study sites that are part of the Enhancing NeuroImaging Genetics through Meta-Analysis (ENIGMA)[61] consortium (13,171 subjects) and Cohorts for Heart and Aging Research in Genomic Epidemiology (CHARGE; 12,803 subjects)[62]. Briefly, the ENIGMA consortium brings together numerous studies, mainly with case–control design, which performed neuroimaging in a range of neuropsychiatric or neuro-degenerative diseases, as well as healthy normative populations. The CHARGE consortium is a collaboration of predominantly population-based cohort studies that investigate the genetic and molecular underpinnings of age-related complex diseases, including those of the brain. An overview of the demographics and type of contribution for each cohort is provided in Supplementary Table 8 (Table adapted from original publication by Adams et al.[42]). Written informed consent was obtained from all participants. The study was approved by the institutional review board of the University of Southern California and the local ethics board of Erasmus MC University Medical Center. Procedures of whole-genome genotyping, imputation, MRI, GWAS, and meta-analysis are summarized in Supplementary Methods (adapted from original publication[42]). The meta-analysis data from the recent ENIGMA2 and CHARGE studies of ICV were available as genome-wide summary statistics, including genome-wide single-nucleotide polymorphism (SNP) data with corresponding $p$-values. The ENIGMA consortium has completed a meta-analysis of site-level GWAS in a discovery sample of 13,171 subjects of European ancestry[2, 61]. Access to the summary statistics of ENIGMA can be requested via their website (http://enigma.ini.usc.edu/download-enigma-gwas-results/). The CHARGE consortium has completed meta-analysis of site-level GWAS in a discovery sample of 12,803 subjects of European ancestry[42]. Genome-wide summary statistics of the CHARGE consortium has been requested by the principal investigator of the study described by Adams et al.[42] For both data sets, only SNPs with an imputation quality score of RSQ ≥ 0.5 and a minor allele frequency ≥0.005 within each site were included.

Procedures of whole-genome genotyping, imputation, magnetic resonance imaging (MRI), GWAS, and meta-analysis of the cohorts are summarized in the Supplementary Methods.

**Gene-based and gene-set analyses**. Gene-based and gene-set analyses were performed using the Multi-marker Analysis of GenoMic Annotation (MAGMA) software package (version 1.02)[63]. First, gene-based $p$-values were calculated using a symmetric 100 kb flanking region for each cohort separately for the 96 autosomal genes in the mTOR pathway. Genome-wide SNP data from a reference panel (1000 Genomes, v3 phase1)[64] was annotated to NCBI Build 37.3 gene locations using a symmetric 100 kb flanking window, and both files were downloaded from http://ctglab.nl/software/magma. Next, the gene annotation file was used to map the genome-wide SNP data from the different studies (ENIGMA2 and CHARGE), to assign SNPs to genes and to calculate gene-based $p$-values for each cohort, separately. Since data from the genome-wide association analyses only included autosomal SNPs, five genes located on the X-chromosome were omitted from the analysis. For the gene-based analyses, single SNP $p$-values within a gene were transformed into a gene-statistic by taking the mean of the $\chi^2$-statistic among the SNPs in each gene. To account for linkage disequilibrium (LD), the 1000 Genomes Project European sample was used as a reference to estimate the LD between SNPs within (the vicinity of) the genes (http://ctglab.nl/software/MAGMA/ref_data/g1000_ceu.zip). Gene-wide $p$-values were converted to $z$-values reflecting the strength of the association of each gene with the phenotype (ICV), with higher $z$-values corresponding to stronger associations. Subsequently, we tested, whether the genes in the mTOR gene-set were jointly associated with ICV in the ENIGMA2 data set, using self-contained and competitive testing[65]. For the gene-set analyses, we used an intercept-only linear regression model including a subvector corresponding to the genes in the gene-set. This self-contained analysis evaluating, whether the regression coefficient of this regression was larger than 0, tests whether the gene-set shows any association with ICV at all. Next, we tested whether genes in each gene-set were more strongly associated with ICV than all other genes in the genome. Therefore, the regression model was then expanded including all genes outside the gene-set. With this competitive test, the differences between the association of the mTOR gene-set to genes outside this gene-set is tested, accounting for the polygenic nature of a complex trait like ICV. To account for the potentially confounding factors of gene size and gene density, both variables as well as their logarithms were included as covariates in the competitive gene-set analysis. Since self-contained tests do not take into account the overall level of association across the genome, gene-size (number of principal components, or SNPs) and gene density, we were interested in the competitive test for the current analysis. The same procedure was followed for analysis of the CHARGE cohort. In addition to

the gene-set analyses within the individual cohorts, we meta-analyzed data of both cohorts on the gene-level followed by gene-set analysis. Post hoc, the potential effects of the two separate mTOR pathways in the gene-set (the PI3K-AKT-mTOR pathway (60 genes) and the RAS-MAPK-mTOR pathway (76 genes)) as well as the individual genes were investigated, by reviewing their gene test-statistics. Moreover, the Reactome_PI3K_AKT_activation gene-set, consisting of 38 genes, was tested for its association with ICV (downloaded from http://software.broadinstitute.org/gsea/msigdb/genesets.jsp). Genes were considered gene-wide significant, if they reached the Bonferroni correction threshold adjusted for the number of genes within the total gene-set ($N = 96$; $p < 0.000521$).

**Generation of zebrafish rheb mutants**. All animal experiments were carried out with the approval of the Institutional Animal Care and Use Committee (IACUC). Guide RNAs targeting the Danio rerio coding region of rheb were generated as described[66]. Subsequently, rheb guide oligonucleotide sequences (rheb_ex3_g1F: 5′-TAGGGTCGTGGAACGCAGCGTTCA-3′ and rheb_ex3_g1R: 5′-AAACT-GAACGCTGCGTTCCACGAC-3′) were ligated into the pT7Cas9sgRNA vector (Addgene) into Bsm BI sites. For the generation of gRNA, the template DNA was linearized with Bam HI, purified by phenol/chloroform extraction and in vitro transcribed using the MEGAshortscript T7 kit (Invitrogen). To generate F0 CRISPR mutants we injected 1 nl containing 100 pg rheb guide RNA and 200 ng Cas9 protein (PNA bio, CP01) to 1-cell stage embryos. To determine the efficiency of the guide RNA, embryos were allowed to grow to 5 days post fertilization (dfp), at which time they were killed and subjected to digestion with proteinase K (Life Technologies) to extract genomic DNA. The targeted locus was PCR amplified using the drrheb_g1test_1 F 5′-GAGTGATCAGCTGTGAAGAAGG-3′ and drrheb_g1test_1 R 5′-GAACAGCGACAGGAGCTACA-3′ primer pair. PCR amplicons were digested using T7 endonuclease I (New England Biolabs) at 37 °C for 1 h and were visualized on a 2% agarose gel. For Sanger sequencing of individual products from the rheb locus, PCR fragments from four embryos with a positive T7assay were cloned into the pCR4/TOPO TA cloning vector (Life Technologies), and 40 colonies from each cloned embryo were Sanger sequenced. We observed sequence aberrations in ~75% of the evaluated rheb clones.

**In vivo modeling in zebrafish embryos**. The human wild-type[67] mRNA of RHEB (NM_005614) was cloned into the pCS2+ vector and transcribed in vitro using the SP6 Message Machine kit (Ambion). The variants identified in RHEB in our patient cohort (RHEBp.P37L, RHEBp.S68P) were introduced using Phusion high-fidelity DNA polymerase (New England Biolabs) and custom-designed primers. We injected 50 pg of WT or mutant RNA into wild-type zebrafish embryos at the 1- to 4-cell stage. For the experiments with rapamycin treatment we added 2.7 nM of ready-made rapamycin solution in DMSO (R8781, Sigma-Aldrich) in each of the injection cocktails. For the headsize assay, the injected larvae were grown to 5 dpf and imaged live on dorsal view. The area of the head was traced excluding the eyes from the measurements and statistical significance was calculated using Student's $t$-test. All experiments were repeated three times and scored blind to injection cocktail.

**Generation of constructs for mouse studies**. The cDNA sequences from human RHEB-WT (NM_005614), and the variants found in the patient cohort (RHEBp.P37L and RHEBp.S68P) were synthesized by GeneCust, and cloned into our dual promoter expression vector. The dual promoter expression vector was generated from the pCMV-tdTomato vector (Clonetech), in which the CMV promoter was replaced with a CAGG promoter followed by a multiple cloning site (MCS) and transcription terminator sequence. To assure expression of the tdTOMATO independent from the gene of interest, a PGK promoter was inserted in front of the tdTomato sequence (for a schematic overview of the expression vector see Supplementary Fig. 3). For all the in vivo and in vitro experiments, the vector without a gene inserted in the MCS was used as control (control vector).

**Mice used for the in vitro and in vivo studies**. For the neuronal cultures, FvB/NHsD females were crossed with FvB/NHsD males (both ordered at 8–10 weeks old from Envigo). For the in utero electroporation female FvB/NHsD (Envigo) were crossed with male C57Bl6/J (ordered at 8–10 weeks old from Charles River). All mice were kept group-housed in IVC cages (Sealsafe 1145T, Tecniplast) with bedding material (Lignocel BK 8/15 from Rettenmayer) on a 12/12 h light/dark cycle in 21 °C (±1 °C), humidity at 40–70% and with food pellets (801727CRM(P) from Special Dietary Service) and water available ad libitum. All animal experiments were approved by the Erasmus MC institutional Animal Care and Ethical Committee, in accordance with European and Institutional Animal Care and Use Committee guidelines.

**In vitro modeling in mouse primary hippocampal neurons**. Primary hippocampal neuronal cultures were prepared from FvB/NHsD wild-type mice according to the procedure described in Goslin and Banker[68]. Briefly, hippocampi were isolated from brains of E16.5 embryos and collected altogether in 10 ml of neurobasal medium (NB, Gibco) on ice. After two washes with NB, the samples were incubated in pre-warmed trypsin/EDTA solution (Invitrogen) at 37 °C for 20 min. After two washes in pre-warmed NB, the cells were resuspended in

1.5 ml NB medium supplemented with 2% B27, 1% penicillin/streptomycin and 1% glutamax (Invitrogen), and dissociated using a 5 ml pipette. Following dissociation, neurons were plated on poly-D-lysine (25 mg/ml, Sigma) coated 15 mm glass coverslips at a density of $3\times10^4$ or $5\times10^4$ cells per coverslip for the axon length measurements and $1\times10^6$ cells per coverslip for all the other experiments. The plates were stored at 37 °C/5% $CO_2$ until the day of transfection. Neurons were transfected at 3 days in vitro (DIV3, DIV7, and DIV14) with the following DNA constructs: control vector (1.8 µg per coverslip), RHEB-WT, RHEBp.P37L, and RHEBp.S68P (all 2.5 µg per coverslip). Plasmids were transfected using Lipofectamine 2000 according to the manufacturer's instructions (Invitrogen). For the rescue experiments, 20 nM Rapamycin (dissolved in 0.01% Ethanol) was applied to the culture 1 day post transfection for 3 days. Neurons were fixed 4 days (Rapamycin experiment) or 5 days (for soma size) post transfection with 4% paraformaldehyde (PFA)/4% sucrose and incubated overnight at 4 °C with primary antibodies in GDB buffer (0.2% BSA, 0.8 M NaCl, 0.5% Triton X-100, 30 mM phosphate buffer, pH 7.4). The following primary antibodies were used: guinea-pig anti MAP2 (1:500, catalogue number: 188004, Synaptic System) to stain dendrites, and rabbit anti-RHEB (1:100, catalogue number: 4935, Cell Signaling). Donkey anti-guinea-pig-Alexa647- and donkey anti-rabbit-Alexa488-conjugated were used as secondary antibodies (all 1:200, catalogue numbers: 706-605-148 and 711-545-152, respectively, Jackson ImmunoResearch). Slides were mounted using mowiol-DABCO mounting medium. Confocal images were acquired using a LSM700 confocal microscope (Zeiss). For the analysis of the neuronal transfections, at least ten distinct confocal images (×20 objective, 0.5 zoom, 1024 × 1024 pixels; neurons were identified by the red immunostaining signal) were taken from each coverslip for each experiment. ImageJ software was used for the analysis of the soma size, by drawing a line around the soma of the cell. For each coverslip, the area of the transfected cells was normalized against the area of the non-transfected cells (five cells per coverslips). These values were then normalized against the mean value of the control (control vector).

### In vivo modeling in mice.
The in utero electroporation was performed as described before[69] Pregnant FvB/NHsD mice at E14.5 of gestation were used to target the progenitor cells giving rise to pyramidal cells of the layer 2/3[70]. The DNA construct (1.5–3 µg/µl) was diluted in fast green (0.05%) and injected in the lateral ventricle of the embryos while still in utero, using a glass pipette controlled by a Picospritzer ® III device. To ensure proper electroporation of the injected DNA constructs (1–2 µl) into the progenitor cells, five electrical square pulses of 45 V with a duration of 50 ms per pulse and 150 ms inter-pulse interval were delivered using tweezer-type electrodes connected to a pulse generator (ECM 830, BTX Harvard Appartus). The positive pole was targeting the developing somatosensory cortex. The following plasmids were injected: control vector, RHEB-WT, RHEBp. P37L and RHEBp.S68P. After birth, pups were sacrificed at P0 or P7 for histochemical processing (to investigate neuronal migration) or used to monitor seizure development.

For the migration analysis, confocal images (×10 objective, 0.5 zoom, 1024 × 1024 pixels) were taken from 2 to 3 non-consecutive sections from 2 and 3 electroporated animals per control and RHEB-containing plasmids, respectively. Images were rotated to correctly position the cortical layers, and the number of cells in different layers were counted using ImageJ using the analyze particles plugin. The results were exported to a spreadsheet for further analysis. Cortical areas from the pia to the ventricle were divided in 10 equal-sized bins and the percentage of tdTOMATO-positive cells per bin was calculated.

For immunofluorescence, mice were deeply anesthetized with an overdose of Nembutal and transcardially perfused with 4% PFA. Brains were extracted and post-fixed in 4% PFA. Brains were then embedded in gelatin and cryoprotected in 30% sucrose in 0.1 M phosphate buffer (PB), frozen on dry ice, and sectioned using a freezing microtome (40/50 µm thick). Free-floating coronal sections were washed in 0.1 M PB and a few selected sections were counterstained with 4′,6-diamidino-2-phenylindole solution (DAPI, 1:10,000, Invitrogen) before being mounted with mowiol® (Sigma-Aldrich) on glass. Overview images of the coronal sections were acquired by tile scan imaging using a LSM700 confocal microscope (Zeiss) with a ×10 objective. Zoom-in images of the targeted area were taken using a ×20 objective. For seizure observations, mice obtained after in utero electroporation were observed daily starting at P18. General behavior was observed by looking for abnormal behaviors such as hyperactivity, the presence of stereotypical behaviors and the presence of tonic–clonic seizures, either spontaneous or induced upon mild handling. Weaned mice were video-monitored for 24 h per day in the Phenotyper (Noldus), to assess seizure onset. Abnormal behaviors and onset of seizures were scored and analysed for each mouse by an expert experimentalist who had been blinded to the identity of samples (i.e., which plasmid had been transfected).

### Statistical analysis used for the mouse studies.
Statistical difference in soma size between the RHEB WT and mutants was determined using one-way analysis of variance (ANOVA) followed by Tukey's post hoc test for multiple comparisons. The effect of Rapamycin treatment on soma size was determined using two-way analysis of variance (ANOVA) followed by Bonferroni's post hoc test for multiple comparisons. For the analysis of the in utero electroporation data, a two-way ANOVA repeated measure was performed, followed by the Bonferroni's multiple

comparisons test. For the analysis of epilepsy onset, the log-rank Mantel–Cox test was used. The significance threshold was set at $p < 0.05$. Data are presented as mean ± standard error of the mean (SEM).

**URLs of used databases**. PubMed (https://www.ncbi.nlm.nih.gov/pubmed/?term=)

EVS database (http://evs.gs.washington.edu/EVS/)
ExAC Browser (query for RHEB) (http://exac.broadinstitute.org/gene/ENSG00000106615)
KEGG database (query for MTOR) (http://www.kegg.jp/kegg-bin/highlight_pathway?scale=1.0&map=map04150&keyword=MTOR)
Ingenuity (http://www.ingenuity.com/)
Access to the summary statistics of ENIGMA2 can be requested via their website: (http://enigma.ini.usc.edu/download-enigma-gwas-results/).

**Data availability**. Material requests for zebrafish experiments should be addressed to N.K. (Nicholas.katsanis@duke.edu). Material requests for neuronal culture and mouse studies should be addressed to G.M.v.W. (g.vanwoerden@erasmusmc.nl) or Y.E. (y.elgersma@erasmusmc.nl). Material requests for gene-set analysis should be addressed to M.Kl. (Marieke.klein@radboudumc.nl) or B.F. (Barbara.Franke@radboudumc.nl). All other material requests should be addressed to M.R. (margot.reijnders@radboudumc.nl) or H.B. (han.brunner@radboudumc.nl).

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

## Acknowledgements

We thank the participating families. We thank all clinicians involved for referring individuals with ID for diagnostic exome sequencing. We thank Hanka Venselaar for interpretation of identified mutations. We thank Francesca Wiersma for technical assistance. The ENIGMA Consortium provided summary statistics of the consortium

findings to this project. The original publication of those findings as well as the list of contributing samples and people can be found on the ENIGMA website: http://enigma.ini.usc.edu/. The Neurology Working Group of the Cohorts for Heart and Aging Research in Genomic Epidemiology (CHARGE) Consortium also contributed with an independent set of summary statistics of consortium findings. The contributing cohorts and persons can be found on previous publications of the CHARGE consortium as listed on the website: http://www.chargeconsortium.com/. Part of this work was carried out on the Dutch national e-infrastructure with the support of SURF Foundation. Barbara Franke is supported by funding from a personal Vici grant of the Netherlands Organisation for Scientific Research (NWO; grant 016-130-669), from the European Community's Seventh Framework Programme (FP7/2007–2013) under grant agreements no. 602805 (Aggressotype) and no. 602450 (IMAGEMEND), and from the European Community's Horizon 2020 Programme (H2020/2014–2020) under grant agreement no. 643051 (MiND). In addition, her work is supported by a grant for the ENIGMA Consortium (grant number U54 EB020403) from the BD2K Initiative of a cross-NIH partnership.

## Author contributions

M.R.F.R. and H.G.B. designed the study. M.R.F.R. and H.G.B. performed cohort analyses. M.Ko., P.L.T., and N.K. performed zebrafish experiments. G.M.v.W., M.P.O., and Y.E. performed studies in neuronal cultures and mice. M.Kl., J.B., and B.F. performed gene-set analyses. A.P.A.S., S.J.C.S., S.H.H., C.G., and R.P. evaluated exome sequencing data. M.R.F.R., T.v.E., E.S., M.v.G., T.K, and H.G.B. collected clinical phenotypes. G.M.M. performed clinical examinations and re-evaluated MRI images. M.R.F.R., M.Ko., G.M.v.W., M.Kl., B.F., Y.E., N.K., and H.G.B., drafted the manuscript. All authors contributed to the final version of the manuscript.

## Additional information

**Competing interests:** B.F. received educational speaking fees from Merz and Shire. The remaining authors declare no competing financial interests.

