## [Peer Review File · Nature Communications]

Reviewer #1 (Remarks to the Author):

This manuscript is exciting as the authors for the first time report mutations in Rheb in patients with ID. This finding adds to the long list of genes mutated in the PI3K-mTOR signaling pathway leading to brain malformations, epilepsy and cognitive deficits. Novelty is essentially in the identification of the mutations. The rest of the data and identified phenotypes are not novel but important to check for the identified mutations. I have the following major comments:

1. Concluding on seizure activity requires EEG recording. It is not sufficient to perform video-monitoring. This section should be amended with either EEG or removed.
2. The recent studies describing a different Rheb mutant 's effect on neuronal migration (Lin et al PNAS 2016) leading to seizures (Hsieh et al Nat comm) should be cited.

minor comments:

1. it is odd to discuss fig. 3 data after fig. 4. Rapamycin may need to be detailed earlier since it is not tested for the in vivo phenotype.

Reviewer #2 (Remarks to the Author):

This is an interesting and timely article on the role of mutations in the MTOR pathway in intellectual disability and brain volume. Mutations in MTOR pathway related genes have been previously described in intellectual disability. Here, based on an astute clinical observation that intellectual disability patients with a high burden of mutations in the MTOR pathway were more likely to have megalencephaly, the authors postulated that the same pathway controls brain volume in the normal population. This is an excellent example of how careful consideration of genetics and phenotype can lead in novel directions. The authors then proceed to prove their hypothesis in large cohorts, and then perform a series of experiments in animal models to underscore their point.

There is little to be added to this already strong article. Though I am not personally a fan of candidate gene studies, or in this case candidate pathway studies, the authors rightly point out that there is pre-existing genetic evidence showing this pathway is involved.

I agree wholeheartedly with paragraph 4 of the discussion in which they point out that the various gene-set databases are incomplete. They have overcome this through their own literature curation, which is reasonable. My only minor quibble on this point is that they seem to imply that nearly every

single gene in this pathway will be important. I doubt that will be the case, and would suspect that it will be only a subset with strong signals.

Reviewer #3 (Remarks to the Author):

In this manuscript Reijnders et al., convincingly demonstrate that MTOR related gene disruptions regulate brain volume in patients with intellectual disabilities. 101 MTOR related genes were evaluated in large ID patient cohort with appropriate controls to identify MTOR activator gene RHEB as a novel ID gene associated with megalencephaly seen in ID patients. Further, various mutant RHEB were shown to affect neuronal cell size, migration, seizure, and head volume in mice and zebrafish. The authors suggest that these changes in neural size and migration may underlie megalencephaly and seizures seen in patients with RHEB mutations. In combination, these studies provide clear evidence for how disrupted MTOR pathway via RHEB affects brain growth in patients with ID. Overall, this is an excellent body of work. The high significance of this work for the field of neurodevelopmental disorders makes this manuscript appropriate for publication in Nature Communications. One minor point that needs some consideration is that the authors suggest increased brain volume in patients may have resulted from an increase in cell size. However, increase in neuronal or glial numbers may also account for large brain volume. At least in P8 brain brains (Figure 4), expression of some of the RHEB mutants may have caused increased number of neurons. This issue should be addressed in the revision.

Point-by-point response to referees' comments

Reviewer #1

1. Concluding on seizure activity requires EEG recording. It is not sufficient to perform video-monitoring. This section should be amended with either EEG or removed.

We agree with the reviewer that EEG recording is the golden standard for measuring seizure *activity*. However, to establish the onset of seizures by 24 hour EEG-recordings in such young pups (P18) is very difficult since the skull is not yet strong enough (and keeps on growing) to carry a pedestal. Hence, our ethical approval does not allow these 24 hour recordings on such young animals. However, the seizures we observed were tonic-clonic seizures, and hence very easy to observe. Other studies in our lab with genes in the mTOR pathway showed that video based scoring of such seizures correlates 100% with EEG-based scoring. Hence we feel confident that the data provided is reliable.

To eliminate confusion, we have now indicated in the text that we specifically scored tonic-clonic seizures and added a supplementary video of one of the pups having such a seizure, to demonstrate that these seizures are very pronounced and can be readily observed.

2. The recent studies describing a different Rheb mutant 's effect on neuronal migration (Lin et al PNAS 2016) leading to seizures (Hsieh et al Nat comm) should be cited.

We added both references to the manuscript. (Page 4, reference numbers 18 and 24)

3. It is odd to discuss fig. 3 data after fig. 4. Rapamycin may need to be detailed earlier since it is not tested for the in vivo phenotype.

We agree with this. In our revised document, we describe the rapamycin experiments (Fig 3) before the paragraph about disturbed neuronal migration (Fig 4). (Page 4)

Reviewer #2

1. I agree wholeheartedly with paragraph 4 of the discussion in which they point out that the various gene-set databases are incomplete. They have overcome this through their own literature curation, which is reasonable. My only minor quibble on this point is that they seem to imply that nearly every single gene in this pathway will be important. I doubt that will be the case, and would suspect that it will be only a subset with strong signals.

We agree with the reviewer that not all genes contribute on a similar level to intracranial volume (ICV). Within our gene set, 18 genes are nominally significant associated with ICV (Supplementary Table 10; Figure S1). We added a sentence to the GWAS results section to emphasize this observation. (Page 3)

Reviewer #3

1. One minor point that needs some consideration is that the authors suggest increased brain volume in patients may have resulted from an increase in cell size. However, increase in neuronal or glial numbers may also account for large brain volume. At least in P8 brain brains (Figure 4), expression of some of the RHEB mutants may have caused increased number of neurons. This issue should be addressed in the revision.

We agree with the reviewer that besides increased soma size, other mechanisms such as increased proliferation or reduced apoptosis, could contribute to a large brain volume. The percentage of neurons that has been targeted with the neuronal migration assay (results shown in Figure 4), only represents 1-10% of the complete somatosensory cortex. Furthermore, the number of targeted cells and the targeted area differs between mice, making any comparison the total amount of targeted cells in absolute numbers between different pups very difficult. Therefore, based on the experiments we performed, we are not able to conclude if *RHEB* mutations cause disturbed neuronal or glial proliferation.

To clarify that other mechanisms than increased soma size can contribute to the observed phenotype, we added a sentence with corresponding reference (reference 27) in the discussion: 'Several mechanisms, such as increased proliferation, increased soma size and reduced apoptosis are known to have a role in the development of megalencephaly.' Afterwards,

we state that '*In vivo*, we postulate that the increased soma size might represent one of the mechanisms through which macrocephaly occurs'. (Page 6)

Reviewer #1 (Remarks to the Author):

The authors still need to take the citation comment seriously. Their introduction and writing in the results section undermine most of the previous studies and basically ignore their findings. In brief, half of the present study is not novel. This does not mean that the present study is not worthy of Nat Comm. The finding of novel mutations and in particular Rheb mutation is of high significance. But this is the only novelty.

They seriously need to acknowledge previous studies related to the function of mTOR signaling pathway including, TSC1, TSC2, and Rheb, on brain development. The previous studies are not limited to cell migration defects but clearly show increased brain size, neuronal dysmorphogenesis, seizures and abnormal behavior. This needs to be clearly cited in the introduction and the results section.

example:

"Previous studies have shown that MTOR signaling is not only involved in cell morphology and growth, but also plays a role in brain development.^{12 176} Increased MTOR activity in vivo, induced either by overexpression of a ¹⁷⁷ constitutively active RHEB or by inactivating mutations in TSC1/TSC2, two negative regulators of RHEB, causes neuronal migration defects.^{16-18.} "

not sufficient!

I strongly advise towards rewriting the introduction and part of the results section and including previous work.

The present work does not even fit the introduction considering that they did not examine ID but only seizures which are not well demonstrated. Clearly EEG could have been done at 4-5 weeks of age. It is not requested at P18. But I am ok to let that go as long as the authors properly acknowledge previous work.

.

Reviewer #3 (Remarks to the Author):

The authors answered my concerns. The manuscript reads very well and brings insightful data supporting the function of MTOR-related genes in brain development and ID.

Point-by-point rebuttal

Reviewer #1 (Remarks to the Author):

The authors still need to take the citation comment seriously. Their introduction and writing in the results section undermine most of the previous studies and basically ignore their findings. In brief, half of the present study is not novel. This does not mean that the present study is not worthy of Nat Comm. The finding of novel mutations and in particularly Rheb mutation is of high significance. But this is the only novelty.

They seriously need to acknowledge previous studies related to the function of mTOR signaling pathway including, TSC1, TSC2, and Rheb, on brain development. The previous studies are not limited to cell migration defects but clearly show increased brain size, neuronal dysmorphogenesis, seizures and abnormal behavior. This needs to be clearly cited in the introduction and the results section.

example:

"Previous studies have shown that MTOR signaling is not only involved in cell morphology and growth, but also plays a role in brain development.^{12 176} Increased MTOR activity in vivo, induced either by overexpression of a ¹⁷⁷ constitutively active RHEB or by inactivating mutations in TSC1/TSC2, two negative regulators of RHEB, causes neuronal migration defects.^{16-18.} "

not sufficient!

I strongly advise towards rewriting the introduction and part of the results section and including previous work.

The present work does not even fit the introduction considering that they did not examine ID but only seizures which are not well demonstrated. Clearly EEG could have been done at 4-5 weeks of age. It is not requested at P18. But I am ok to let that go as long as the authors properly acknowledge previous work.

According to the reviewer and editors request, we revised the Introduction and parts of the Results and Discussion. We added to the manuscript (marked with comments in the main text):

Introduction:

- One additional paragraph, with appropriate references (References 4-35). (Page 3)

Results:

- Text in "Rapamycin rescues neuronal soma and head size defects" and References 33-35 (Page 5)

- Text in "RHEB mutations affect neuronal migration and induce seizures" and References 12-14 (Pages 5 and 6)

Discussion:

- We removed reference Takei et al (2014), since it refers to a Review. (Page 7)

- Text and References 32, 6,11,15, 52, 53, and 33-35. (Page 7 and 8)